# Facilitation of Hand Proprioceptive Processing in Paraplegic Individuals with Long-Term Wheelchair Sports Training

**DOI:** 10.3390/brainsci12101295

**Published:** 2022-09-26

**Authors:** Tomoyo Morita, Eiichi Naito

**Affiliations:** 1Center for Information and Neural Networks (CiNet), Advanced ICT Research Institute, National Institute of Information and Communications Technology (NICT), Osaka 565-0871, Japan; 2Graduate School of Frontier Biosciences, Osaka University, Osaka 565-0871, Japan

**Keywords:** functional magnetic resonance imaging, hand movement, proprioceptive processing, primary motor cortex, intraparietal sulcus region, right frontoparietal cortices, wheelchair sports, paraplegia

## Abstract

Previous studies have revealed drastic changes in motor processing in individuals with congenital or acquired limb deficiencies and dysfunction. However, little is known about whether their brains also exhibit characteristic proprioceptive processing. Using functional magnetic resonance imaging, we examined the brain activity characteristics of four individuals with congenital or acquired paraplegia (paraplegic group) who underwent long-term wheelchair sports training, when they passively experienced a right-hand movement (passive task) and when they actively performed a right-hand motor task (active task), compared to 37 able-bodied individuals (control group). Compared with the control group, the paraplegic group showed significantly greater activity in the foot section of the left primary motor cortex and in the inferior frontoparietal proprioceptive network during the passive task. In the paraplegic group, the left intraparietal sulcus region was activated during the passive task, but suppressed during the active task, which was not observed in the control group. This shows the facilitation of hand proprioceptive processing and unique usage of the intraparietal sulcus region in proprioceptive motor processing in the brains of paraplegic individuals with long-term wheelchair sports training.

## 1. Introduction

Previous studies on the motor processing in individuals with congenital or acquired limb deficiencies and dysfunction revealed drastic changes [1,2,3,4,5,6,7,8]. Yet, little is known about the characteristic proprioceptive processing that their brains exhibit.

Somatotopic representation is likely different in the brains of individuals with congenital or acquired limb deficiencies and dysfunction. For example, the hand sections of the primary somatosensory (SI) and motor (M1) cortices [6,8] and the cerebellum [7] have shown to be activated during foot motor tasks in individuals with congenital hand deficiencies. Similarly, the foot section of M1 is reportedly activated during a bimanual motor task in some individuals with congenital or acquired paraplegia (dysfunction of bilateral lower limbs) [9]. These findings suggest that the somatotopic sections of a missing or dysfunctional limb can be replaced by another limb in core sensorimotor brain regions that have clear and distinct gross somatotopy. However, since motor tasks were used in these studies, it remains unclear whether such replaced somatotopic sections are solely involved in the proprioceptive processing of a replacing limb.

In the present study, we addressed this question by examining the brain activity characteristics of four individuals with congenital or acquired paraplegia who underwent long-term wheelchair sports training (paraplegic group; Table 1) using functional magnetic resonance imaging (fMRI). The subjects passively experienced a right-hand movement (passive task) and actively performed a right-hand motor task (active task), and the data were compared to those of 37 able-bodied individuals (control group).

A previous study showed that the hand or foot section of M1 is activated during the proprioceptive processing of each limb in able-bodied individuals [10]. We hypothesized that the foot section could be activated during the proprioceptive processing of the hand in the paraplegic group if the foot section of M1 in this group were involved in the proprioceptive processing of the hand. Furthermore, the foot section of M1 is reportedly deactivated during a manual motor task in young able-bodied participants [9,11]. Thus, in the present control group, it was likely for the foot section to not be activated during proprioceptive processing of the hand. We then examined whether activity in the foot section of M1 during the passive task was greater in the paraplegic group than in the control group, and checked the activity increase in each paraplegic participant.

The finding that the foot section was replaced by the hand in some paraplegic participants [9] suggests that neuronal resources in their foot section additionally participated in the information processing of the hand. This implies that the brains of paraplegic individuals may have the potential to assign greater neuronal resources for hand information processing than those of able-bodied individuals. Moreover, paraplegic individuals likely have received a greater amount of proprioceptive input from their hands through their daily manual handling of a wheelchair, leading to the facilitation of proprioceptive processing of the hand. Hence, we may assume that, in addition to the foot section of M1, other brain regions of the paraplegic group as well would show greater activity associated with the hand-related proprioceptive processing than those in the control group. The regions involved in the higher-order proprioceptive network, such as the right inferior parietal lobule (IPL), right inferior frontal gyrus (IFG, especially area 44), and the bilateral insula cortices [10,12,13,14,15,16], are shown to be activated during proprioceptive processing of the right hand in able-bodied individuals. Importantly, these regions have local hand and foot sections that are activated during proprioceptive processing of the left and right hands and feet with considerable overlap between these sections [10]. Hence, we hypothesized that activity of the higher-order proprioceptive network during the passive task in the paraplegic group would be greater than in the control group. We also tested this hypothesis by comparing brain activity during the passive tasks in the paraplegic and control groups.

Finally, if proprioceptive processing of the hand were facilitated in the paraplegic group, we would expect distinct proprioceptive and motor processing in their brains. Therefore, we also explored brain regions that showed paraplegic group-specific differences between the passive and active tasks.

## 2. Materials and Methods

### 2.1. Participants

We recruited one participant with congenital paraplegia (P1) and three participants with acquired paraplegia (P2, P3, and P4) as paraplegic participants. Details of the participants are presented in Table 1. P1, aged 30 years, was an active, top wheelchair-racing Paralympian. She began racing at the age of eight and underwent long-term wheelchair racing training using the bilateral upper limbs. However, P2, P3, and P4 were not top athletes like P1. They had more than 30 years of leg non-use and long-term wheelchair sports training (Table 1). P1, P3, and P4 had no somatic sensations (light touch and pinprick) in their lower limbs which had complete immobility. P2 had complete immobility of his lower limbs; nonetheless, there were somatic sensations (light touch and pinprick). These were evaluated by a physiotherapist with more than 10 years of experience. The handedness of the participants was confirmed using the Edinburgh Handedness Inventory [17]. P1, P3, and P4 were right-handed, and P2 was ambidextrous (Table 1).

For the control group, we recruited right-handed able-bodied adults (n = 37; 25 females) aged 37.4 ± 10.9 years (range, 25–59 years). They had experience in various sports since their school days; however, none were athletes participating in a particular sport. Their ages were somewhat lower than those of the paraplegic participants, although neither paraplegic participant’s age was significantly different from the control group, as revealed by a Crawford and Howell *t*-test (P1, *p* > 1; P2, *p* = 0.08; P3, *p* = 0.29; P4, *p* = 0.40 after Bonferroni correction) [18]. The handedness of the control participants was confirmed using the Edinburgh Handedness Inventory (95.9 ± 7.6). None of the participants had a history of neurological, psychiatric, or movement disorders.

The Ethics Committee of the National Institute of Information and Communications Technology (NICT) and the magnetic resonance imaging (MRI) Safety Committee of the Center for Information and Neural Networks (CiNet; no. 2003260010) approved this study. The details of the experiment were explained to each participant before the experiment, after which they provided their written informed consent. This study was conducted following the principles and guidelines of the Declaration of Helsinki (1975).

### 2.2. General Procedure

We reported fMRI results when the paraplegic and able-bodied groups performed passive and active tasks of right-hand movement. The results of both groups performing a bimanual motor task have been previously reported [9].

Before the fMRI experiment, we explained the tasks to be performed in the scanner to every participant. To familiarize themselves with the tasks, they performed the tasks outside the MRI room. They then entered the room and were placed in an MRI scanner. Their heads were immobilized using sponge cushions and adhesive tape and their ears were plugged. Their body parts (chest, pelvis, and shin) were fixed to the MRI bed using Velcro to reduce body movements during the task. When performing a task, they were asked to close their eyes, relax their entire body, refrain from producing unnecessary movements, and think of only the assigned task.

Each participant completed one experimental 160 s run for each task. The run comprised five task epochs, each lasting 15 s. The details of each task are described below. The task epochs were separated by 15 s baseline (rest) periods. Each run also included a 25 s baseline period before the start of the first epoch. During the experimental run, participants were provided with auditory instructions that indicated the start of a task epoch (i.e., three, two, one, start). A “stop” instruction generated by a computer was also provided to notify the participants of the end of each epoch. All auditory stimuli were provided using MRI-compatible headphones. An experimenter who stood beside the scanner bed checked whether the participants were performing each task properly through visual inspection throughout the run.

### 2.3. MRI Data Acquisition

Functional MRI images were acquired using T2*-weighted gradient echo-planar imaging (EPI) sequences with a 3.0-Tesla MRI scanner (MAGNETOM Trio Tim; Siemens, Germany) and a 32-channel array coil. A multiband imaging technique was used (multiband factor = 3) [19]. Each volume consisted of 48 slices (slice thickness = 3.0 mm) which were acquired in an interleaved manner to cover the entire brain. The repetition time (TR) was 1000 ms. We used an echo time (TE) of 27 ms and a flip angle (FA) of 60°. The field of view was 192 × 192 mm and the matrix size was 64 × 64 pixels. The voxel dimensions were 3 × 3 × 3 mm along the *x*, *y*, and *z* axes, respectively. In total, 160 volumes were collected for each experimental run.

A T1-weighted magnetization-prepared rapid gradient echo image was also acquired using the same scanner for each participant, which was used in the subsequent image preprocessing stage. The imaging parameters were as follows: TR = 1900 ms, TE = 2.48 ms; FA = 9°, field of view 256 × 256 mm^2^, matrix size 256 × 256 pixels, slice thickness = 1.0 mm; voxel size 1 × 1 × 1 mm^3^, and 208 contiguous transverse slices.

### 2.4. Passive Task

In each task epoch, an experimenter (EN) continuously generated cyclic extension–flexion movements of the participants’ right wrist in synchronization with 1 Hz cyclic tones. We prepared a device to control the range of wrist motion (Figure 1A) that was used in a previous study [20]. A movable hand rest was mounted on the device and the hand was fixed on it, indicating the wrist angle. Two stoppers were fixed onto the device to control the range of wrist motion across task epochs and participants. They were positioned to prevent the wrist from extending beyond a straight position (0°) and flexing beyond 60°. The participants’ right wrists were fixed on this device, and the experimenter controlled the continuous wrist extension–flexion movements from 0° to 60°. The participants were asked to relax their right hand and passively experience movements without generating them. During the passive task, the brain most likely receives and processes somatic inputs from various receptors in the muscles, joints, and skin that signal right-hand movement [15]. Thus, this task is suitable for evaluating how the brain processes proprioceptive inputs from the right hand.

### 2.5. Active Task

In each task epoch, all participants continuously exerted cyclic extension–flexion movements of their right wrist in synchronization with 1 Hz cyclic tones by themselves. Participants’ right wrists were fixed to the same device (Figure 1B). The participants had to control continuous wrist extension–flexion movements while touching one of the stoppers (0° or 60°) alternately with the hand-rest in synchronization with the 1 Hz audio tones. An example of kinematic data when performing this task was presented in our previous study [20].

### 2.6. fMRI Data Preprocessing and Single-Subject Analysis

To eliminate the effects of unsteady magnetization, we discarded the first 10 EPI images from each fMRI run. The imaging data were analyzed using SPM 12 (Wellcome Centre for Human Neuroimaging, London, UK) implemented in MATLAB (MathWorks, Sherborn, MA, USA). We performed the following preprocessing steps for each participant using the SPM default parameters. First, all EPI images were aligned to the first EPI image of the first session with six degrees of freedom (translation and rotation about the *x*, *y*, *z* axes) of rigid displacement. Using this procedure, we obtained data related to the position of the head, which changed over time from the first frame. All participants had a maximum displacement of less than 1.5 mm in the *x*, *y*, or *z* plane and less than 3° of angular rotation about each axis during each fMRI run. Therefore, we excluded no data from the analysis. For each participant, the T1-weighted structural image was co-registered to the mean image of all realigned EPI images using affine transformation. The structural and realigned EPI images were spatially normalized to the standard stereotactic Montreal Neurological Institute (MNI) space [21]. Finally, the normalized images were spatially smoothed using a Gaussian kernel with a full width at half a maximum of 4 mm along the *x*, *y*, and *z* axes.

After preprocessing, we used a general linear model [22,23] to analyze the fMRI data. A design matrix was prepared for each participant. As for this single-subject analysis, the design matrix contained a boxcar function for the task epoch in the run, which was convolved with a canonical hemodynamic response function. Six realignment parameters were also included in the design matrix (regressors of no interest) to correct the residual motion-related variance after realignment. In the analysis, global mean scaling was not performed to avoid inducing type I errors in the evaluation of negative blood oxygenation level-dependent responses (deactivation) [24]. We generated an image showing the task-related activity in each task for each participant, which was used in subsequent analyses. In this image, the effect of cyclic tones was most likely eliminated as the participants heard the sound consistently during the task epochs and rest periods.

### 2.7. Brain Regions Active during Each Task in the Control Group

To depict the brain regions that were active during the passive and active tasks in the entire brain, we first identified the regions that were significantly activated during each task in the control group by performing a second-level group analysis [25] using a parametric one-sample *t*-test (Figure 1). We also examined brain regions that showed task differences (active > passive, passive > active) in this group (Figure 1). For the contrast active > passive, we used an image defined by active > baseline (uncorrected height threshold of *p* < 0.05) as an inclusive mask. Using the mask image, we ensured that the observed activation was true activation during the active task rather than pseudo-activation caused merely by deactivation during the passive task. The same procedure was used for the contrast passive > active. We used the family-wise error rate (FWE)-corrected extent threshold (*p* < 0.05) for the entire brain for a voxel-cluster image generated at an uncorrected height threshold of *p* < 0.005. We did not perform a parametric analysis in the paraplegic group because of the small number of participants.

### 2.8. Nonparametric Test to Evaluate Group Differences

To examine group differences, we used permutation-based statistical nonparametric mapping (SnPM). We used a non-parametric method because the number of paraplegic participants was too small to guarantee a normal distribution of the data obtained from this group, and the variance of the data in each group was unlikely to be the same between the paraplegic and control groups. We performed a nonparametric two-sample *t*-test implemented in the SnPM13 toolbox (RRID: SCR_002092) [26], with default settings, 10,000 permutations, and no variance smoothing. In this analysis, age, sex, and handedness were included as nuisance covariates. We used this nonparametric test in the following whole-brain and ROI analyses to investigate group differences in each task and in the exploration of paraplegic group-specific differences between passive and active tasks.

### 2.9. Evaluation of Group Differences in the Whole Brain

We first identified brain regions in the whole brain that showed significantly greater activity in the paraplegic group than in the control group for each task, using the nonparametric test described above. We used the FWE-corrected extent threshold (*p* < 0.05) for the entire brain for voxel-cluster images generated at an uncorrected height threshold of *p* < 0.005. Whole-brain analysis revealed that the paraplegic group had significant clusters of voxels showing greater activity in the left medial wall motor region, right IPL, bilateral IFG, and insula than the control group during the passive task (Figure 2 and Table 2). In contrast, none of the regions showed significantly greater activity in the paraplegic group during the active task. As we found significant clusters in the entire brain during the passive task, we further performed the following ROI analyses to test our hypotheses.

### 2.10. Group Difference in the Foot Section of the M1

To check if the cluster in the left medial wall motor region identified in the whole brain analysis (Figure 2) was located in the foot section of the left M1, we performed an ROI analysis. We defined the foot section of M1 based on the data obtained from our previous study [10], in which we identified brain regions upon vibrating the tendon of the right foot, while 19 able-bodied participants experienced an illusory movement of their right foot. We defined the foot section of the left M1 (M1 ROI; pink section in Figure 3A) by depicting the overlapping region between the functional cluster in the left medial wall region identified in the previous study and the cytoarchitectonic maps for areas 4a and 4p, implemented in the JuBrain Anatomy toolbox v. 3.0 [27]. We performed a nonparametric two-sample *t*-test and searched for significant clusters of voxels in the M1 ROI. We generated a voxel-cluster image at an uncorrected height threshold of *p* < 0.005, and checked for significant clusters in the ROI by small volume collection (SVC) [28].

We identified a significant cluster of voxels in the M1 ROI during the passive task (green section in Figure 3A). To check the activity of the cluster during each task in either group, we extracted parameter estimates from the cluster for each participant and calculated the mean value of the parameter estimates across participants for each task in each group (Figure 3B).

Figure 3B shows that the significant group differences in the passive task were likely due to increased activity in the paraplegic group and decreased activity in the control group. To evaluate the significance of the activity decrease (compared with zero) in the control group, we conducted a parametric one-sample *t*-test using the parameter estimate obtained from the ROI. This was also performed for the active task. In the paraplegic group, a one-sample *t*-test could not be used due to the small number of participants. Therefore, we simply described the activity of each paraplegic participant.

### 2.11. Group Difference in the Higher-Order Proprioceptive Network

To check if the clusters in the right IPL and the bilateral IFG and insula identified in the whole brain analysis (Figure 2) were located in the higher-order proprioceptive network, we performed another ROI analysis. Four ROIs were defined based on the data obtained from a previous study [10]. We first depicted the inferior frontoparietal cortices that were active during illusions of all limbs (functional clusters). We then defined the IPL and area 44 ROIs (pink section in Figure 4A) by depicting the overlapped region between the functional clusters and the cytoarchitectonic maps for the right areas PF, PFt, PFm, PFcm, and PFop (IPL ROI), and those for the right area 44 (Area 44 ROI) implemented in the JuBrain Anatomy toolbox v. 3.0. For the insula ROIs, since the cytoarchitectonic maps for the insular region in the JuBrain Anatomy toolbox are still limited to the anterior part, we defined the left and right insula ROIs (pink section in Figure 4A) by depicting the overlapped region between the functional clusters and the insula region implemented in the Automated Anatomical Labeling (AAL) atlas [29]. We performed a nonparametric two-sample *t*-test and searched for significant clusters of voxels in each ROI. We generated a voxel-cluster image generated at an uncorrected height threshold of *p* < 0.005 and checked significant clusters in each ROI with SVC.

We identified a significant cluster of voxels in each of the IPL and area 44 ROIs, as well as two significant clusters of voxels in each of the left and right insula ROIs during the passive task (green section in Figure 4A). To check the locations of these clusters relative to the regions active during the passive task in the control group (Figure 1C), we superimposed the clusters and active regions onto the MNI standard brain (Figure 4B).

Finally, to check the activity of the cluster(s) in each ROI during each task in each group, we extracted parameter estimates from the cluster(s) for each participant. Here, we extracted parameter estimates from the cluster in each IPL and area 44 ROIs, and from the two clusters in each of the left and right insula ROIs. We then calculated the mean value of the parameter estimate across participants for each task in each group (Figure 4C–F).

### 2.12. Nonparametric Test to Explore Paraplegic Group-Specific Difference between Passive and Active Tasks

Finally, we explored the brain regions that showed paraplegic group-specific differences between passive and active tasks (paraplegic group (passive–active)–control group (passive–active)). We used a test (two-group, two-condition test) implemented in SnPM13. In this analysis, we identified a significant cluster of active voxels in the left intraparietal sulcus (IPS) region (Figure 5A). As we found three peaks in this region, we extracted parameter estimates from a 4 mm radius sphere around each peak for each participant and calculated the mean value of parameter estimates across participants for each task in each group (Figure 5B). We also examined the opposite contrast (control group (passive–active)–paraplegic group (passive–active)).

## 3. Results

### 3.1. Brain Regions Active during Passive and Active Tasks in the Control Group

In the control group, the passive task significantly activated the hand sections of the left primary sensorimotor cortices including the dorsal premotor cortex, bilateral supplementary motor areas (SMA), IPL, parietal operculum, IFG, insula, frontal operculum, thalamus, putamen, and cerebellum (Figure 1C and Appendix A). Highly similar regions were activated during the active task (Figure 1D). As for task differences, the active task significantly activated the right cerebellar vermis and hemisphere more strongly than in the passive task (Figure 1E). Similarly, a cluster with a significant trend was observed in the hand section of the left primary motor cortex (M1), including the dorsal premotor cortex (*p* = 0.07). In the control group, none of the regions was significantly more strongly activated during the passive task than during the active task. The results are summarized in Appendix A.

### 3.2. Group Difference in the Whole Brain

In the passive task, the paraplegic group showed significant clusters of voxels with greater activity in the left medial wall motor region, right IPL, bilateral IFG, and insula than the control group (Figure 2 and Table 2). In contrast, none of the regions showed significantly greater activity in the paraplegic group during the active task. Similarly, none of the regions showed significantly greater activity in the control group for either task.

### 3.3. Group Difference in the Foot Section of the Left M1

We examined whether the cluster in the left medial wall motor region identified during the passive task in the whole-brain analysis (Figure 2) was located in the M1 ROI. We found a significant cluster in the ROI (49 voxels, peak coordinates = −4, −26, 68; Figure 3A).

In this cluster (green section in Figure 3A), we found increased activity in the paraplegic group and decreased activity in the control group during the passive task (Figure 3B). Although activity increase was a group effect in the paraplegic group, activity seemed to increase in P1, P3, and P4, but not in P2 (Figure 3B). In contrast, in the control group, we confirmed a significant decrease in activity from zero during the passive task by using a parametric one-sample t-test (Figure 3B).

We also found decreased activity in the cluster (green section in Figure 3A) in the control group during the active task, and confirmed a significant decrease in activity from zero by conducting a one-sample t-test. During the active task, we found larger inter-participant variability in the paraplegic group; that is, P1 showed a robust activity increase, whereas P2 showed a robust activity decrease.

Taken together, the foot section of the left M1, which was suppressed during the proprioceptive processing (passive task) and active control (active task) of the right hand in the control group seemed to have been involved in the proprioceptive processing of the right hand in the paraplegic group (in at least three of the four paraplegic participants).

### 3.4. Group Difference in the Higher-Order Proprioceptive Network

We examined whether the activity in the right IPL and bilateral IFG and insula identified during the passive task in the whole-brain analysis (Figure 2) was located in the ROIs of the higher-order proprioceptive network (pink sections in Figure 4A). We found a significant cluster in every IPL (156 voxels) and area 44 ROI (61 voxels). Two significant clusters were identified in the ROI of the right insula. One was in the anterior insula (155 voxels) and the other was in the middle insula (93 voxels). Similarly, we also found two significant clusters in the ROI of the left insula. One was located in the anterior insula (48 voxels), and the other in the middle insula (20 voxels). Hence, the activity of the higher-order proprioceptive network was augmented during proprioceptive processing of the right hand in the paraplegic group compared with the activity in the control group.

The significant clusters identified in the IPL and area 44 ROIs (green sections in the left two panels in Figure 4B) were located posteriorly to the regions that were active during the passive task in the control group (red sections in the left two panels in Figure 4B), with some overlap (yellow sections in the left two panels in Figure 4B). Similarly, significant bilateral anterior insular clusters (green sections in the two right panels in Figure 4B) were located anterior to the regions activated during the passive task in the control group (red sections in the two right panels in Figure 4B). These results suggest that the regions of the higher-order proprioceptive network, in which the paraplegic group showed significantly greater activity during the passive task, included regions that were not active during the passive task in the control group (Figure 4B).

In the cluster(s) identified in each ROI, greater activity was observed during the passive task in the paraplegic group than in the control group (Figure 4C–F), verifying the results of the ROI analysis (Figure 4A). In the paraplegic group, although the group effect showed an increase in activity during the passive task, the activity increase in P2 (green dots) was less than that observed in the other paraplegic participants in the clusters of all ROIs (Figure 4C–F).

Among the cluster(s) identified in each ROI, no robust group differences were observed during the active task in area 44 and bilateral insula clusters (Figure 4D–F). In contrast, the paraplegic group showed greater activity in the IPL cluster than the control group.

### 3.5. Paraplegic Group-Specific Difference between Passive and Active Tasks

We found a significant cluster in the left IPS (Figure 5A). This region was not significantly activated during either the passive or active tasks in the control group (Figure 1). The peaks in this cluster are summarized in Table 3. Three peaks were observed in the different cytoarchitectonic regions. When we examined the activity of each peak for each task and group, we found a highly similar pattern of activity across all three peaks (Figure 5B). At all peaks, we found increased activity during the passive task and decreased activity during the active task in the paraplegic group, which was not observed in the control group (Figure 5B). Finally, no regions were identified in the opposite contrast (control group (passive–active)–paraplegic group (passive–active)).

## 4. Discussion

### 4.1. Limitation in the Present Study

The current study has some limitations. Only a limited number of paraplegic participants were recruited owing to the restrictions imposed by COVID-19, although a relatively large number of able-bodied control participants was recruited. However, (1) increased activity in the foot section of the left M1 during the passive task (Figure 3), (2) increased activity in the higher-order proprioceptive network during the passive task (Figure 4), and (3) increased activity during the passive task and decreased activity during the active task in the left IPS (Figure 5) were observed in at least three out of four paraplegic participants, suggesting the general consistency of these findings across the present paraplegic participants.

### 4.2. Proprioceptive Processing in the Foot Section of the Left M1 in the Paraplegic Group

A significant decrease in activity in the foot section of the left M1 during the passive and active tasks in the control group (Figure 3B) can be considered as cross-somatotopic inhibition [30,31], in which the brain tries to suppress the occurrence of an unintended foot movement during hand movement. In contrast, in the paraplegic group, the activity in the M1 cluster increased during the passive task across participants, except for P2 (Figure 3B). We may point out the difference in P2 from the remaining participants in that he had somatic sensations (light touch and pinprick) in his lower limbs (Table 1). Therefore, it was possible that his foot section of M1 was still being used for the information processing of the foot (see also Section 4.2), although he also had complete immobility of the lower limbs, similar to the other paraplegic participants. This view seems to be corroborated by the finding that P2 showed a decrease in activity in the left M1 cluster during the active task, which was the pattern observed in the control group (Figure 3B). Altogether, the present paraplegic participants (P1, P3, and P4) with no somatic sensations from the lower limbs consistently used the foot section of the left M1 for proprioceptive processing of the right hand.

In the paraplegic group, the group effect was an increase in the activity in the foot section of the left M1 (M1 ROI) during the passive task, whereas no increase was observed during the active task (Figure 3B). This indicates that the use of the foot section for proprioceptive processing of the hand does not necessarily imply that this section is used for active motor control of the hand. The participants differed in many factors (Table 1). Some of these factors could be related to the larger inter-participant variability in the use of the foot section for active motor control (Figure 3B).

In contrast, the relatively consistent use of this section for hand proprioceptive processing compared with hand motor control in the present paraplegic participants (Figure 3B) suggests that the use of the foot section of M1 for hand proprioceptive processing could be a more general phenomenon than the use of this section for hand motor control. A previous study [9] has proposed that the use of the foot section of M1 for information processing of the hand could be better referred to as hyperadaptation. The current results imply the possibility that such M1 hyperadaptation starts with proprioceptive function rather than motor function.

In a previous study, in which we reported individual activity in the foot section of the left M1 during a bimanual motor task in the present paraplegic participants [9], significant activation was reported only in P1 and P4, but not in P2 and P3. As shown in the previous study [9], the foot section did not always show significant activity during a unimanual active task in all paraplegic participants (Figure 3B). Furthermore, the previous study [9] reported the specificity of P1 in terms of the activation and structural changes in the foot section. Consistent with this finding, the strongest activation during the passive and active tasks was observed in P1 among all paraplegic participants (Figure 3B).

### 4.3. Facilitation of Activity in the Higher-Order Proprioceptive Network in the Paraplegic Group

As hypothesized, the paraplegic group showed significantly more activity in the ROIs of the higher-order proprioceptive network during the passive task than the able-bodied group (Figure 4). Together with the activity increase in the foot section of M1 during the passive task (Figure 3), the series of results indicated the facilitation of hand proprioceptive processing in the paraplegic group.

The paraplegic group showed significantly greater activity not only in the active regions (yellow sections in Figure 4B) but also in those that were not active (green sections in Figure 4B) during the passive task in the control group. Thus, the latter regions, which were not used for hand proprioceptive processing in the control group, were additionally used for processing in the paraplegic group. Local somatotopic organization (hand/arm and foot/leg sections) may exist in the IPL, IFG, and insula of higher-order proprioceptive networks in humans, with considerable overlap between these sections [10,32,33,34]. Hence, it is likely that the local foot sections in these regions additionally participated in hand proprioceptive processing in the paraplegic group, contributing to their activity augmentation in these regions.

In the clusters of all ROIs (Figure 4C–F), P2 showed a relatively smaller increase in activity during the passive task, as observed in the M1 cluster (Figure 3B). As discussed above, the relatively smaller activity increase in higher-order proprioceptive processing could be explained by the fact that the brain does not fully allocate neural resources for foot information processing to hand information processing because the foot sections in the higher-order proprioceptive network are still being used for original information processing of the foot.

We observed activation in the right IPL and area 44 during the passive task, even in the control group (Figure 1 and Figure 4B). These regions are known to be connected by the inferior branch of the superior longitudinal fasciculus [12], and it has been consistently shown that these regions in the right hemisphere are coactivated during proprioceptive processing of the hand in able-bodied participants (Figure 1). The exact roles of these regions in proprioceptive processing remain unclear. However, a previous study has shown that activity in the right IPL and areas 44/45 are well correlated with the amount of proprioceptive illusory movement of the right hand experienced by able-bodied participants [12]. Thus, activity in these regions is likely associated with proprioceptive awareness; that is, the somatic sensation of the hand moving.

In the passive task, the paraplegic group also showed significantly greater activity in the bilateral insular cortices than the control group (Figure 4). In general, the insula is the main locus of the salience network, which involves bottom-up detection of salient events [35,36]. In the present study, during the passive task, we found bilateral insular activation in both the control (Figure 1) and paraplegic (Figure 4B) groups. This indicates that the bilateral insular cortices are involved in the bottom-up detection of proprioceptive events (passive hand movement). However, it was only in the paraplegic group that activity increased in insular regions where clusters were found during the passive task (Figure 4E,F). This suggests that the response of these insula regions was specific to the paraplegic group. The greater activity in the insular clusters during the passive task in this group suggests that the paraplegic group had a higher sensitivity to detecting passive hand movement and that hand movement was more noticeable due to the facilitation of hand proprioceptive processing. This knowledge would provide valuable clinical information for understanding body perception in paraplegic individuals.

### 4.4. Role of the IPS Region in the Paraplegic Group

In the left IPS region, activity increased during the passive task and decreased during the active task in the paraplegic group (Figure 5B–D). This is a novel finding of this study. This activity pattern was distinct from that observed in the high-order proprioceptive network, where the paraplegic group showed significantly greater activity during the passive task than did the control group (Figure 4C–F), suggesting the function of the IPS region to be different from that of the higher-order proprioceptive network. Moreover, since this region was consistently silent during the passive task in the control group (Figure 1C and Figure 5B–D), the function of the IPS was likely unique to the paraplegic group.

When we applied a lenient extent threshold (voxel > 150) in the exploration of brain regions showing paraplegic group-specific differences between the passive and active tasks (see Section 2.12), we found additional clusters in the left pre-SMA and bilateral inferior frontal cortices, including the anterior insula (see Appendix A, Appendix A and Appendix A). These regions showed a similar pattern of brain activity (see Appendix A–D) to that observed in the left IPS (Figure 5B–D), suggesting that the left IPS worked concertedly with these brain regions during the passive and active tasks in the paraplegic group.

The IPS, bilateral inferior frontal cortices including the anterior insula and pre-SMA are the main constituents of a network involved in response inhibition, which has been consistently reported in go/no-go and stop-signal tasks [37,38,39,40]. In particular, it was recently shown that the IPS functionally connects with the inferior frontal cortices and pre-SMA, and that the IPS plays a role in stopping the occurrence of action [40].

In the passive task, we asked the participants to relax their right hand to avoid generating actual movements. In the brains of the paraplegic group, in which hand proprioceptive processing was facilitated (Figure 2, Figure 3 and Figure 4), the likelihood of automatic generation of motor commands from proprioceptive signals was likely to increase [15]. Therefore, for paraplegic participants to relax their right hand during the passive task, their brains could have activated the IPS of the response inhibition network to actively suppress the possible occurrence of hand movement. Conversely, during the active task in which the participants had to move their hands, this network may have been suppressed. The paraplegic individuals would have proprioceptive processing facilitated by long-term experiences of handling a wheelchair and may have developed unique neuronal circuits that suppress the possible occurrence of involuntary movements when their hands are passively moved by the wheelchair.

These are our speculations, and further investigations would unveil the unique role of the IPS during hand proprioceptive motor processing in the brains of paraplegic individuals. However, if our view is correct, the current work could be the first to demonstrate the recruitment of a response inhibition network by an internal drive during passive movements in the brains with proprioceptive processing facilitated (i.e., the brains having neuronal states where an actual movement is easier to generate during passive movement task).

## Figures and Tables

**Figure 1 brainsci-12-01295-f001:**
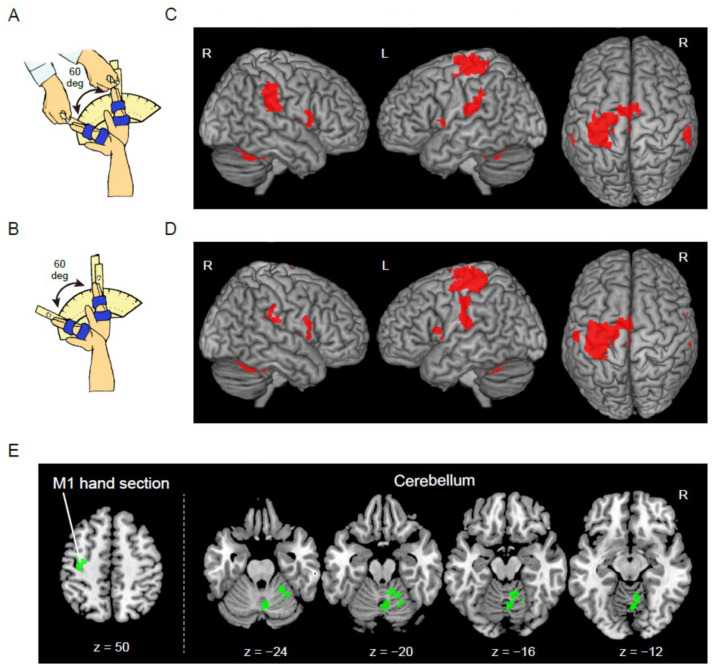
Functional magnetic resonance imaging (fMRI) tasks and results of the control group. (**A**) Passive task. (**B**) Active task. (**C**) Brain regions active during the passive task. (**D**) Brain regions active during the active task. These sections are superimposed on the MNI standard brain. The right hemisphere, left hemisphere, and the top view are shown in each panel. (**E**) Brain regions (M1 hand section and cerebellum) where the control group showed greater activity during the active than during the passive task. These regions are superimposed on the transverse slices (z = 50, −24, −20, −16, and −12) of the MNI standard brain. Abbreviations: R, right; L, left; M1, primary motor cortex; MNI, Montreal Neurological Institute.

**Figure 2 brainsci-12-01295-f002:**
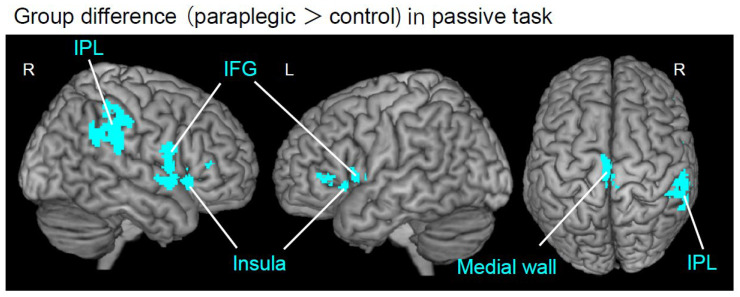
Results from whole brain analysis. Brain regions (cyan sections) showing significant group difference during the passive task are superimposed on the MNI brain. The paraplegic group showed significant clusters of voxels having greater activity in the left medial-wall motor region, right IPL, bilateral IFG, and insula than the control group. The right hemisphere, left hemisphere, and top view are shown in each panel. Abbreviations: R, right; L, left; IPL, inferior parietal lobule; IFG, inferior frontal gyrus; MNI, Montreal Neurological Institute.

**Figure 3 brainsci-12-01295-f003:**
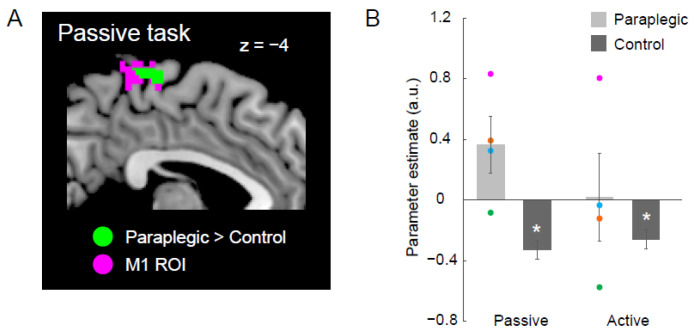
Results from the ROI analysis for M1. (**A**) Significant cluster in which the paraplegic group showed greater activity during the passive task than the control group (green section) within the M1 ROI (pink section). The cluster and the ROI are superimposed on the sagittal slice (x= −4) of the MNI brain. (**B**) Bar graphs show mean value of brain activity (parameter estimate) obtained from the identified M1 cluster across participants for each task in both the paraplegic group (light gray) and control group (dark gray). Individual data from the paraplegic group are also plotted (pink, green, orange, and blue circles represent data obtained from P1, P2, P3, and P4, respectively). Lines on the bars indicate standard errors of mean across participants. Asterisks indicate a significant decrease of activity in the control group (*p* < 0.001). Abbreviations: a.u., arbitrary unit; M1, primary motor cortex; ROI, region-of-interest.

**Figure 4 brainsci-12-01295-f004:**
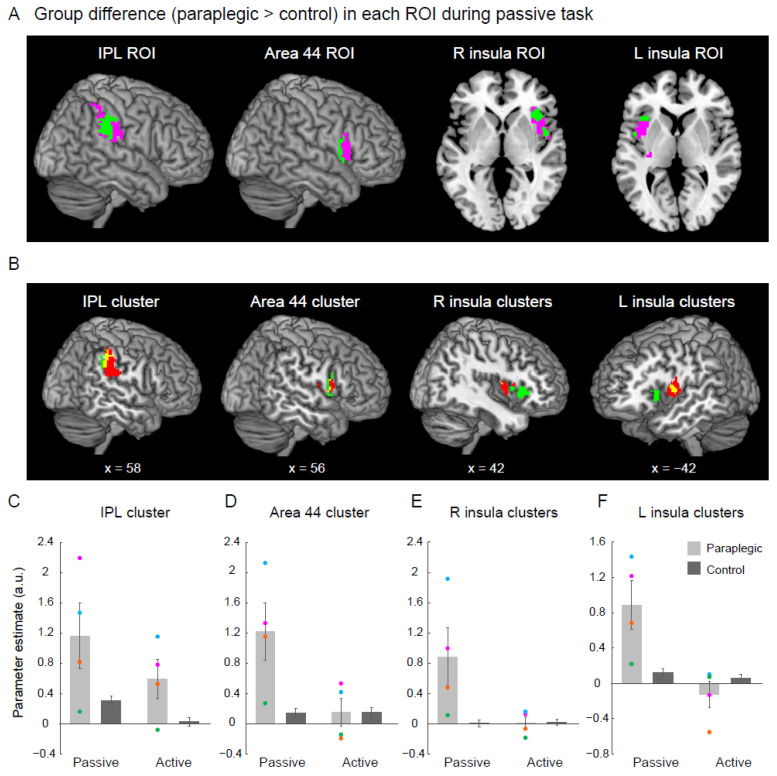
Results from the ROI analysis for the higher-order proprioceptive network. (**A**) Significant cluster(s) in which the paraplegic group showed greater activity during the passive task than the control group (green section) within the IPL, area 44, R insula, and L insula ROIs (pink sections). The clusters and the ROIs are superimposed on the MNI brain. (**B**) The significant clusters identified in each ROI (green sections) and the brain regions active during the passive task in the control group (red sections) are superimposed on the MNI brain. Yellow sections indicated the overlapping areas of the two sections. (**C**–**F**) Bar graphs show the mean value of brain activity (parameter estimate) obtained from the identified cluster(s) in each ROI across participants for each task in both paraplegic group (light gray) and control group (dark gray). Individual data from the paraplegic group are also plotted (pink, green, orange, and blue circles represent the data obtained from P1, P2, P3, and P4, respectively). Lines on the bars indicate standard errors of the mean across participants. Abbreviations: a.u., arbitrary unit; R, right; L, left; IPL, inferior parietal lobule; IFG, inferior frontal gyrus; MNI, Montreal Neurological Institute; ROI, region of interest.

**Figure 5 brainsci-12-01295-f005:**
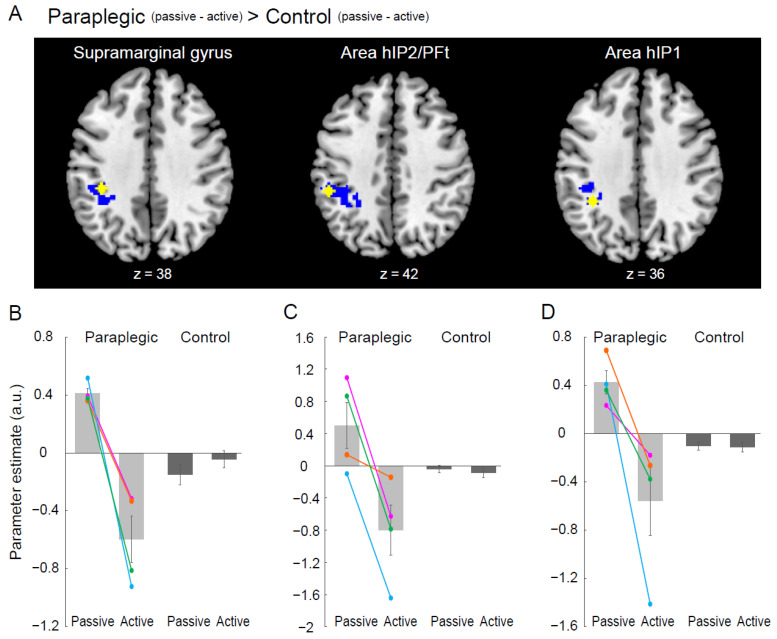
(**A**) The left IPS region showing a significant paraplegic group-specific difference between the passive and active tasks (paraplegic group (passive–active)–control group (passive–active)). The identified IPS region has three peaks in the supramarginal gyrus, area hIP2/PFt, and area hIP1, respectively. The IPS region (blue section) is superimposed on the transverse slices of the MNI brain (*x* = 38, 42, and 36), with each peak plotted as a yellow dot. (**B**–**D**) Bar graphs show mean value of brain activity (parameter estimate) of each peak across participants for each task (yellow in each of paraplegic group (light gray) and control group (dark gray). Individual data from the paraplegic group are also plotted (pink, green, orange, and blue circles and lines represent the data obtained from P1, P2, P3, and P4, respectively). Lines on the bars indicate standard errors of the mean across participants. Abbreviations: a.u., arbitrary unit; IPS, intraparietal sulcus; MNI, Montreal Neurological Institute.

**Table 1 brainsci-12-01295-t001:** Participants’ information.

**Participant**	**P1**	**P2**	**P3**	**P4**
Age (in years)	30	61	54	52
Sex	F	M	M	M
Leg non-use period	30 years	60 years	37 years	31 years
Neurological level	T12	Cannot be specified	T3	T8
Cause	* Spina bifida	Poliomyelitis	Spinal cord injury	Spinal cord injury
ASIA impairment scale	A	undefined	A	A
SCI	Complete	undefined	Complete	Complete
FIM	101/126	108/126	101/126	101/126
Somatic sensations (light touch and pin prick) from lower limbs	No	Yes	No	No
Wheelchair sports (years played) Training period (age), training days/week, training hours/day	Track racing and marathon (23) 8–14 yo, 2/w, 8 h 15–17 yo, 7/w, 1–8 h 18–30 yo, 6/w, 2 h	Basketball (42) 17–22 yo, 4–5/w, 3 h 23–58 yo, 1–2/w, 3 h Table tennis (4) 15–18 yo, 1/w, 2 h	Table tennis (27) 28–54 yo, 2–4 w, 2.5–8 h	Basketball (31) 22–35 yo, 5/w, 3 h 36–52 yo, 1/w, 3 h Marathon (9) 27–35 yo, 2/w, 3 h Fencing (9) 27–35 yo, 2/w, 3 h
Handedness score	60	7	100	90

Note: * Congenital, ASIA: American Spinal Injury Association, SCI: Spinal Cord Injury, FIM: Functional Independence Measurement, T: thoracic.

**Table 2 brainsci-12-01295-t002:** Brain regions that showed significant group difference in each task identified in whole brain analysis.

Cluster	Size (voxels)	*x*	*y*	*z*	*t*-Value	Anatomical Identification
Passive (paraplegic − control)						
R IPL	492 *	62	−34	34	7.16	Area PF
		62	−32	20	5.26	IPL
		54	−30	46	5.00	Area PFt
L IFG/insula	456 *	−36	12	14	7.02	Area OP8
		−54	12	2	4.41	Area 44
R IFG/insula	448 *	54	6	8	7.00	Area 44
		52	4	−6	6.65	Insula
L medial-wall motor	382 *	−6	−14	66	6.67	Area 6mc (SMA)
		−4	−22	56	5.02	Area 4a
R middle IFG/insula	398 *	48	22	−6	6.37	IFG
		38	20	2	5.34	Frontal operculum cortex
Active (paraplegic − control)						
*No significant cluster*						

Height threshold, *p* < 0.005 uncorrected; extent threshold, *p* < 0.05, FWE-corrected across the entire brain. For the anatomical identification of peaks, we only considered cytoarchitectonic areas available in the anatomy toolbox that had a probability greater than 30%. The cytoarchitectonic area with the highest probability is reported for each peak. When cytoarchitectonic areas with > 30% probability were not available to determine a peak, we simply provided the anatomical location of the peak. In each cluster, we reported peaks that were more than 8 mm apart from each other in the order of larger *t*-values. Asterisks indicate statistical significance of *p*-value corrected for entire brain volume (*, *p* < 0.05). R, right; L, left; IPL, inferior parietal lobule; IFG, inferior frontal gyrus.

**Table 3 brainsci-12-01295-t003:** Brain regions showing paraplegic group-specific difference between passive and active tasks.

Cluster	Size (voxels)	*x*	*y*	*z*	*t*-Value	Anatomical Identification
Paraplegic (passive–active)–control (passive–active)						
L IPS	643 *	−38	−34	38	5.04	Supramarginal gyrus
		−50	−36	42	4.98	Area hIP2/PFt
		−36	−44	36	4.94	Area hIP1

Asterisks indicate statistical significance of *p*-value corrected for entire brain volume (*, *p* < 0.05). L, left; IPS, intraparietal sulcus.

## Data Availability

The data supporting the findings of this study are available upon request from the corresponding author. The data are not publicly available because they contain information that can compromise the privacy of the research participants.

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
