# Peer review of "Facilitation of Hand Proprioceptive Processing in Paraplegic Individuals with Long-Term Wheelchair Sports Training"

_brainsci, 2022, doi:10.3390/brainsci12101295_

Round 1
Reviewer 1 Report
The manuscript examines the proprioceptive processing in individual with paraplegia either acquired or congenital through fMRI. it found abnormal pattern of activation of motor areas and left intraparietal sulcus compared with healthy people. I think that it is important to understand the sensory impairments and rehabilitation and management of sensory impairments. Please find my following comments;
1- Overall, the manuscript needs to be rewritten in shorter and informative way. It is very long and hard to follow.
2- Rewrite the introduction. It is not true to start with the description of experiments.
3- Mention in the introduction the current evidence, limitation , and what your study can add
4- what is the reference value in the one-sample t-test
5- The discussion re-mentioned the study results without sufficient explanation
6- There is no clinical or research recommendation based on the study results
Author Response
We appreciate you taking the time to offer us your comments on our work.
1- Overall, the manuscript needs to be rewritten in more consious and informative way.
Response:
We appreciate this comment. However, since no other reviewers have indicated any serious problems for this point, we would like to keep the present format.
2- Rewrite the introduction. It is not true to start with the description of experments.
Response:
According to the reviewer’s comment, we have moved the description of the experiments lightly back in the introduction.
3- Mention in the introduction the current evidence, limitation, and what your study can add
Response:
In the introduction, we have already described the current evidence and limitation in the previous studies, and what our study can add to the previous knowledge, so we would like to keep it in the present format.
4- what is the reference value in the one-sample t-test
Response:
In the Methods and Results sections (section 2.10 and section 3.3) of the manuscript (brainsci-1878851), we added information that reference value for the one-sample t-test was zero. Please see lines 328, 402, and 406.
5- The discussion re-mentioned the study results without sufficient explanation
Response:
As many researchers do, we provided a summary of the three main results at the beginning of the discussion, and each result was discussed in the following sections (sections 4.1, 4.2, and 4.3, respectively). We think this style would assist the readers’ understanding.
6- There is no clinical or research recommendation based on the study results
Response:
In the last part of the Discussion (section 4.2) of the manuscript (brainsci-1878851), we have already mentioned “the paraplegic group has a higher sensitivity to detect passive hand movement and that hand movement is more noticeable due to the facilitation of hand proprioceptive processing.” in the discussion (lines 722-723 This was clinically important information. According to the reviewer’s comment, we have further added the sentence “This knowledge would provide valuable clinical information for understanding body perception in paraplegic people.” in the manuscript (brainsci-1878851) (lines 724-725 section 4.2).
Reviewer 2 Report
The authors have revised the manuscript and the queries have been addressed. The author has removed and modified the phrase "our previous study" in the article. However, the phrase has been retained in many other places. Hence to maintain uniformity the author is advised to modify the rest of the manuscript in lines numbered - 183, 245, 647, 652, 656, 690
Author Response
The authors have revised the manuscript and the queries have been addressed. The author has removed and modified the phrase "our previous study" in the article. However, the phrase has been retained in many other places. Hence to maintain uniformity the author is advised to modify the rest of the manuscript in lines numbered - 183, 245, 647, 652, 656, 690
Response:
We appreciate this comment. We have modified these parts. Please see our revised manuscript.
Round 2
Reviewer 1 Report
The authors did not address the required comment.
The manuscript is still need a lot of modification in writing style and English language especially introduction and discussion.
What is the reference value for one-sample t-test?
Author Response
We appreciate this reviewer for taking his/her time for our work.
Point1
The manuscript is still need a lot of modification in writing style and English language especially introduction and discussion.
Response:
According to this request, we have made a substantial modification especially in Introduction and Discussion to reduce redundancy. The revised manuscript has been edited by professional English editors (Editage, www.editage.com).
Point2
What is the reference value for one-sample t-test?
Response:
We think we have already answered this question. The reference value for the one-sample t-test was zero. In the Methods and Results sections (section 2.10 and section 3.3) of the revised manuscript, we added this information. If this is not the correct answer, could you please tell us what information you need.